# Code sets for respiratory symptoms in electronic health records research: a systematic review protocol

Wikum Jayatunga,[1] Philip Stone,[2] Robert W Aldridge,[1] Jennifer K Quint,[3] Julie George[1]

[1]Institute of Health Informatics, University College London, London, UK
[2]National Heart and Lung Institute, Imperial College London, London, UK
[3]Respiratory Epidemiology, Occupational Medicine and Public Health, Imperial College London, London, UK

**Correspondence to**
Dr Wikum Jayatunga;
wikumj@gmail.com

## ABSTRACT

**Introduction** Asthma and chronic obstructive pulmonary disease (COPD) are common respiratory conditions, which result in significant morbidity worldwide. These conditions are associated with a range of non-specific symptoms, which in themselves are a target for health research. Such research is increasingly being conducted using electronic health records (EHRs), but computable phenotype definitions, in the form of code sets or code lists, are required to extract structured data from these large routine databases in a systematic and reproducible way. The aim of this protocol is to specify a systematic review to identify code sets for respiratory symptoms in EHRs research.

**Methods and analysis** MEDLINE and Embase databases will be searched using terms relating to EHRs, respiratory symptoms and use of code sets. The search will cover all English-language studies in these databases between January 1990 and December 2017. Two reviewers will independently screen identified studies for inclusion, and key data will be extracted into a uniform table, facilitating cross-comparison of codes used. Disagreements between the reviewers will be adjudicated by a third reviewer. This protocol has been produced in accordance with the Preferred Reporting Items for Systematic Reviews and Meta-Analyses Protocol guidelines.

**Ethics and dissemination** As a review of previously published studies, no ethical approval is required. The results of this review will be submitted to a peer-reviewed journal for publication and can be used in future research into respiratory symptoms that uses electronic healthcare databases.

**PROSPERO registration number** CRD42018100830.

## Strengths and limitations of this study

► To the best of our knowledge, this is the first systematic review to identify code sets for respiratory symptoms used in electronic health records research.

► The results will allow future researchers of respiratory symptoms to reuse or build on already established code sets, which is particularly relevant to health services research that may be more concerned with symptom presentation than diagnosis.

► Due to poor reporting of code sets in the published literature, we may identify studies that have used a code set without including further information about it either in their main paper or in supplementary material.

► Code sets derived from different databases and coding classifications may be difficult to compare and this heterogeneity may limit our ability to synthesise results that are generalisable.

## INTRODUCTION

Respiratory illnesses, such as asthma and chronic obstructive pulmonary disease (COPD), result in significant morbidity worldwide.[1] There is also known to be a large degree of underdiagnoses, as these conditions are common and the initial symptoms are mild, such that patients may not present to primary care or receive a definitive diagnosis in the early stages of disease.[2] The most common symptoms associated with these conditions include breathlessness, wheeze, coughing and sputum production.[3 4] These symptoms are relatively non-specific, being associated with other medical conditions, such as heart failure or pneumonia, which creates further diagnostic challenges.[5] Thus, research into symptom presentation is important, as it could capture those patients presenting for care who may not have received a formal diagnosis.

The widespread use of electronic health records (EHRs) in healthcare has resulted in the creation of large population-based databases, which present unique opportunities for research.[6] Research using EHRs typically begins with the extraction of structured data: clinical symptoms, diagnoses and treatments, which have been recorded in databases as clinical codes.[7] Given the range and variety of codes included in classification systems, a collection or list of unique values ('code set' or 'code list') can be used to identify a particular clinical concept. These code sets can be used, sometimes alongside other data, such as medications and test results, to define the computable phenotype and comprehensively

capture all patients with a variable of interest.[8 9] For more complex clinical concepts, logical expressions may be needed to combine codes; but for simpler concepts, a simple list of relevant codes may be used. Defining this list of codes is a key early step in the research process, as the omission of important codes and inclusion of inappropriate codes can lead to selection biases, affecting the numbers and types of patients captured for study, which in turn impacts on the validity of research findings.[7 10] However, as EHR research is a relatively new field, code sets are often poorly reported in published studies, as are the methods used to generate them.[10]

In recent years, best practice guidelines have begun to be established to improve the validity and reproducibility of research using EHRs.[11 12] For example, methods have been developed describing how to construct code sets in a more systematic and reproducible way[13 14] and online repositories have been developed to aid in the sharing of code sets.[15] A recent review identified different methods of code set development, such as creating a list of synonyms for the condition of interest, use of hierarchies in coding classification systems, reviewing code sets with clinician input and updating existing code sets.[8] The publication of code sets, and the process of their construction, ensures that research conducted using EHRs is transparent, reproducible and can be scrutinised.

Previous studies have aimed to establish phenotyping methodologies and validation for conditions, such as asthma[16] and COPD.[17 18] However, to the best of our knowledge, there is currently no current consensus or systematic review around code lists for respiratory symptoms. This is particularly relevant for health services research and system planning, which may be more concerned with patient presentation than formal diagnosis. A review of existing code sets for respiratory symptoms would also be useful for future researchers, potentially allowing them to reuse or build on already established code sets.

## METHODS AND ANALYSIS

The primary objective of this review is to develop an understanding of the code sets used for identifying respiratory symptoms in EHR research. Relevant questions include:

1. What code sets have been developed for looking at respiratory symptoms from EHRs?
2. How do these code sets compare in terms of the type, length and breadth of codes included?
3. How are code sets combined to define particular symptoms?

### Search strategy

A comprehensive search of the MEDLINE and Embase databases (via the HDAS interface) will be conducted. A search strategy has been developed using a combination of keywords and MeSH (Medical Subject Heading) terms relating to (1) EHRs, (2) code sets and (3) respiratory symptoms, including synonyms for these terms.

The full search strategy with terms used can be found in online supplementary file 1. Additionally, the reference lists of articles found will be reviewed for other potential studies, and where the code set itself is not fully reported in a paper, we will contact the authors of the paper to obtain this. In addition, the following code set repositories will be searched for information existing outside of the peer-reviewed literature: ClinicalCodes. org,[19] CALIBERcodelists,[20] CPRD@Cambridge Codelists,[21] Phenotype Knowledge Base,[22] the NIH Value Set Authority Centre[23] and the LSHTM Data Compass.[24] The study screening process will be recorded using the Preferred Reporting Items for Systematic Reviews and Meta-Analyses (PRISMA) flow diagram.[25]

### Eligibility criteria

The inclusion criteria are: studies looking at respiratory symptoms (specifically: cough, shortness of breath, wheeze and sputum) that are based on research using routine data from EHRs, and that describe the use of a code set (or equivalent). Broader search terms for 'respiratory symptoms' are included in the search strategy to improve its sensitivity, but after full-text screening, only papers relating to the above respiratory symptoms will be included. All potential coding classification systems will be eligible. Studies in the English language from any country and published between 01 January 1990 and 31 December 2017 will be included. Exclusion criteria are: studies with code sets for respiratory diagnoses only, rather than symptoms, and studies based on primary data collection rather than secondary data analysis of routine data from electronic databases.

### Data collection

Identified studies will be stored using Rayyan systematic review software,[26] and duplicate studies will be removed. Two investigators will separately assess the title and abstracts against the eligibility criteria to screen for papers for inclusion. For papers, where either reviewer is unclear whether it meets the eligibility criteria, the full paper will be screened. Eligible papers will then have full-text screened, and disagreements about inclusion will be resolved by discussion with a third reviewer to reach a consensus. Key data from each included study will be extracted and recorded in a uniform table using Microsoft Excel. The key data variables that will be extracted are: study details (title, first author and year of publication); population (country, time period and target population); research question or area of interest; target symptom or symptoms; data source (type of healthcare database); code classification system; version of the code classification system; code set; coding algorithm for symptom ascertainment and method used to develop code set. The quality of code set development methodology will be scored using methods established in the literature as a benchmark.[13 14] The code sets are the main data item of interest, and they will be summarised and tabulated

to facilitate cross-comparison of the codes included in each list. A narrative synthesis[27] will be conducted to describe differences in the code sets.

## Limitations

It is known that code sets are poorly reported in the published literature.[10] Therefore, we may identify studies that appear to have used a code set but then do not report further information about how it was developed or sourced. Furthermore, including a search term relating to 'code sets' may not pick up papers that have not reported the code set development in their main paper or relegated this information to an appendix. However, including this search term was necessary to improve the specificity of the search. Another limitation is that code sets derived from different systems and using different classification systems may limit our ability for cross-comparison and synthesis of results. Finally, as this is a retrospective review of published code sets, the results may become outdated as clinical terminologies are updated over time, and so researchers using the results from this review to inform their work should be aware of this and conduct additional searches for any new clinical codes of relevance.

## Bias

As individual study outcomes are not of interest in this systematic review, we will not consider the risk of bias in individual studies. A publication bias may exist as there may be studies using code sets that are not identified as they have not been published in the literature (eg, studies that did not have positive findings), though this will not be formally assessed.

We used the PRISMA-Protocol checklist[28] when writing this protocol (see online supplementary file 2).

## Patient and public involvement

Patients and the public were not involved in the development of this protocol.

## ETHICS AND DISSEMINATION

This review synthesises information that is already in the public domain, and does not directly involve human participants, and so ethical approval is not required. Findings of the review will be disseminated via presentation at relevant conferences and publication in a peer-reviewed journal.

**Contributors** JG and JKQ developed the research question. WJ summarised the literature and drafted the protocol, with input into the methodology, search strategy and manuscript from PS, JKQ, JG and RWA. All authors read, commented on and approved the final manuscript. Any amendments or updates to this protocol will be documented by WJ.

**Funding** JG is funded by a Health Education England/National Institute of Health Research Clinical Lectureship (ICA-CL-2016-02-024). RWA is supported by a Wellcome Trust Clinical Research Career Development Fellowship (206602/Z/17/Z).

**Competing interests** None declared.

**Patient consent for publication** Not required.

**Provenance and peer review** Not commissioned; externally peer reviewed.

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
