## [Reviewer comments · BMJ Open]

ARTICLE DETAILS

TITLE (PROVISIONAL)	Code sets for respiratory symptoms in electronic health records (EHR) research: a systematic review protocol
AUTHORS	Jayatunga, Wikum; Stone, Philip; Aldridge, Robert; Quint, Jennifer; George, Julie

VERSION 1 - REVIEW

REVIEWER	Mohammad Al Sallakh Swansea University Medical School
REVIEW RETURNED	22-Aug-2018

GENERAL COMMENTS	This is a well written protocol to survey codelists (or code sets) of respiratory symptoms (and signs?) in the EHR literature, which will inform the wider respiratory research and help researchers capture undiagnosed patients. I have very minor comments: 1. In Page 6, Line 42, the authors wrote "The inclusion criteria are: studies looking at respiratory symptoms (specifically: cough, shortness of breath, wheeze and sputum) ...". The word "specifically" conveys that the Authors will only focus on those symptoms, whereas the rules 8 and 11 of the search strategy include any upper or lower respiratory signs and symptoms.2. In the title, depending on #1, the Authors may wish to change "symptoms" to "signs and symptoms" or "observations".3. In Page 6, Line 20, I suggest replacing "specific symptoms" with "particular symptoms" (if that what the Authors mean), since "specific symptoms" may be confused with the opposite of "non-specific symptoms".4. Page 7, Line 16. In EHR/database research, I understand that "coding algorithms" sometimes refers to algorithms to ascertain health conditions using clinical codes (and thus it is included in the search strategy). However, I think the term should more appropriately refer to algorithms that codify narrative clinical notes (e.g., using NLP). In this manuscript, I would use something like "symptom ascertainment algorithm" instead.5. I recommend using the term "code set" (or "codeset") instead of "code list"/"codelist" throughout the protocol and the systematic review. This is also the recommendation #12 by Williams et al 2017 (reference [8] in this protocol).
--

	In data and computer science, a list is an ordered collection of values which allows duplicates, while a set is an unordered collection of unique values (see https://en.wikipedia.org/wiki/List_(abstract_data_type) , https://en.wikipedia.org/wiki/Set_(abstract_data_type) , and https://stackoverflow.com/q/1035008). In clinical code sets, there is (and should be) no meaning of the order of values (e.g., for cough, 171A., 171F., 1715. is the same as 1715., 171A., 171F.). The values may be ordered in some way just for convenience. However, the database query (typically) doesn't consider that order in selecting the matching health event records.
--	---

REVIEWER	Susan H. Fenton UT School of Biomedical Informatics
REVIEW RETURNED	07-Oct-2018

GENERAL COMMENTS	Interestingly, codelist is never formally defined. Is it truly just a list of codes? What about other types of identifiers, such as concept identifiers? The definition in the CALIBER codelist documentation is more complex than a simple list of codes. The introduction appears to indicate that only a codelist is needed for a computable phenotype. Generally, computable phenotypes consist of a variety of data, for example, medications, diagnostic test results and more. The last sentence in the second paragraph of the introduction is confusing. Diagnostic and procedural indices were utilized prior to EHRs. Why would use of an EHR impact the codelist?
--

REVIEWER	Jessica Watson University of Bristol UK
REVIEW RETURNED	18-Oct-2018

GENERAL COMMENTS	Overall this is a well written paper addressing an important area of improving transparency and rigor in codelists for Electronic Health Record research. Issues: 1) My main concern relates to the search strategy. Using a term relating to codelists limits the papers to those specifically looking at codelist development. Most papers using electronic health records will only report these codelists as a very small part of the methods section - these papers will therefore not be picked up on this search, although they may include codelists as an appendix. Researchers outside of the respiratory field (in particular cancer diagnostics) will also use relevant codelists for respiratory symptoms which will not be picked up by this search strategy. Consider modifying the search strategy or addressing this in the limitations section. 2) Another limitation is that whilst diagnoses tend to be well defined (eg using ICD-10), symptoms are often less clearly defined. Different studies may therefore use different inclusion/exclusion criteria which may make it more difficult to synthesise or cross compare. Would be helpful to more clearly define your symptoms of interest a priori eg using International Classification of Primary Care (ICPC).
---

	3) Although bias in the studies may not be an outcome of interest I would like to see how you plan to address the quality of the methods used for developing the codelists, eg by using previously published methods as a benchmark (such as Watson et al 2017 https://bmjopen.bmj.com/content/7/11/e019637) 3) Finally I would like to see acknowledgement in the limitations section that using a retrospective review of published codelists inevitably means that codelists produced will be outdated. This is particularly important given migration to SNOMED-CT. As a result researchers should supplement pre-existing codelists with additional searches for any new clinical codes of relevance. 4) Finally please include the PROSPERO registration number.
--	--

REVIEWER	Alison K Wright University of Manchester, UK
REVIEW RETURNED	26-Oct-2018

GENERAL COMMENTS	This protocol reports on the criteria for a systematic review of code lists for respiratory symptoms used in electronic health record research. This is an important first step when designing and developing a study using EHR data and ensures research conducted is reproducible and more comparable between databases. The authors have developed an extensive code list search strategy and identified a good range of code list repositories to search for articles not found through their literature search. Another repository the authors may wish to consider is the LSHTM Data Compass (http://datacompass.lshtm.ac.uk/) which provides code lists for research produced by LSHTM and its collaborators. A minor point, in the EMBASE and Medline search strategy provided in the supplement, the time window for consideration is reported as 1990-2017 whereas in the protocol it is stated as 1990-2018.
--

VERSION 1 – AUTHOR RESPONSE

REVIEWER: 1

This is a well written protocol to survey codelists (or code sets) of respiratory symptoms (and signs?) in the EHR literature, which will inform the wider respiratory research and help researchers capture undiagnosed patients. I have very minor comments:

1.1

In Page 6, Line 42, the authors wrote "The inclusion criteria are: studies looking at respiratory symptoms (specifically: cough, shortness of breath, wheeze and sputum) ...". The word "specifically" conveys that the Authors will only focus on those symptoms, whereas the rules 8 and 11 of the search strategy include any upper or lower respiratory signs and symptoms.

We are specifically interested in the respiratory symptoms described on page 22 lines 51-52. Rules 8 and 11 of the search strategy on page 13, which relate more broadly to "respiratory symptoms" are

included to improve the sensitivity of the search and ensure studies are not missed, but after full text screening, only papers relating to the respiratory symptoms described will be included. This has now been clarified on page 22, lines 53-57.

1.2

In the title, depending on #1, the Authors may wish to change "symptoms" to "signs and symptoms" or "observations".

We are specifically interested in respiratory symptoms in this protocol, not signs. Rule 8 (on page 13) in the search strategy, which refers to "Signs and Symptoms, Respiratory" is a Medical Subject Heading (MeSH) term designed to capture all potentially relevant studies that have been indexed under that topic. However, we will only be including papers with code sets for the respiratory symptoms described on page 22 lines 51-52.

1.3

In Page 6, Line 20, I suggest replacing "specific symptoms" with "particular symptoms" (if that what the Authors mean), since "specific symptoms" may be confused with the opposite of "non-specific symptoms".

We have amended the above as suggested, replacing the word "specific" with "particular" on page 22, line 28.

1.4

Page 7, Line 16. In EHR/database research, I understand that "coding algorithms" sometimes refers to algorithms to ascertain health conditions using clinical codes (and thus it is included in the search strategy). However, I think the term should more appropriately refer to algorithms that codify narrative clinical notes (e.g., using NLP). In this manuscript, I would use something like "symptom ascertainment algorithm" instead.

We have amended the above on Page 23, Lines 22-24, replacing "coding algorithms" with "coding algorithm for ascertaining symptoms" which we believe communicates that we are referring to the algorithms used to ascertain symptoms via clinical codes.

1.5

I recommend using the term "code set" (or "codeset") instead of "code list"/"codelist" throughout the protocol and the systematic review. This is also the recommendation #12 by Williams et al 2017 (reference [8] in this protocol). In data and computer science, a list is an ordered collection of values which allows duplicates, while a set is an unordered collection of unique values (see [https://en.wikipedia.org/wiki/List_\(abstract_data_type\)](https://en.wikipedia.org/wiki/List_(abstract_data_type)) , [https://en.wikipedia.org/wiki/Set_\(abstract_data_type\)](https://en.wikipedia.org/wiki/Set_(abstract_data_type)) , and <https://stackoverflow.com/q/1035008>).

In clinical code sets, there is (and should be) no meaning of the order of values (e.g., for cough, 171A., 171F., 1715. is the same as 1715., 171A., 171F.). The values may be ordered in some way just for convenience. However, the database query (typically) doesn't consider that order in selecting the matching health event records.

We have amended the above as suggested by the reviewer, changing all references of "code list" in the manuscript to "code set". Where 'code set' is first mentioned (Page 21, Line 28) we also write "or code list" in recognition of the fact that some will be more familiar with the latter term. 'Code list' is still included as a synonym of 'code set' in the search strategy to improve the sensitivity of the search and capture papers that may have only used the former term.

REVIEWER: 2

2.1

Interestingly, codelist is never formally defined. Is it truly just a list of codes? What about other types of identifiers, such as concept identifiers? The definition in the CALIBER codelist documentation is more complex than a simple list of codes.

We have reworded the section where code sets/lists are introduced to include a definition and clearer description of the term (Page 21, Lines 27-34).

2.2

The introduction appears to indicate that only a codelist is needed for a computable phenotype. Generally, computable phenotypes consist of a variety of data, for example, medications, diagnostic test results and more.

On Page 21, Lines 31-32, we have added the phrase “sometimes alongside other data such as medications and test results” to reflect the comment that other data is often used alongside a code set to define a computable phenotype.

2.3

The last sentence in the second paragraph of the introduction is confusing. Diagnostic and procedural indices were utilized prior to EHRs. Why would use of an EHR impact the codelist?

Apologies for the confusion. Our point is not that the codes themselves are impacted by EHR use, but that there has been poor reporting of code sets in published research papers where EHRs have been used. To clarify our meaning we have added the phrase “code sets are often poorly reported in published studies” Page 21, Line 42.

REVIEWER: 3

Overall this is a well written paper addressing an important area of improving transparency and rigor in codelists for Electronic Health Record research. Issues:

3.1

My main concern relates to the search strategy. Using a term relating to codelists limits the papers to those specifically looking at codelist development. Most papers using electronic health records will only report these codelists as a very small part of the methods section - these papers will therefore not be picked up on this search, although they may include codelists as an appendix. Researchers outside of the respiratory field (in particular cancer diagnostics) will also use relevant codelists for respiratory symptoms which will not be picked up by this search strategy. Consider modifying the search strategy or addressing this in the limitations section.

We have added a new limitation section which addresses the above comment (Page 23, Lines 33-41). It has been necessary to include a search term relating to “codelist/set” to improve the specificity of the search. Without such a term included, the search strategy yields over 42,000 results, a number which would not be practical to screen.

3.2

Another limitation is that whilst diagnoses tend to be well defined (eg using ICD-10), symptoms are often less clearly defined. Different studies may therefore use different inclusion/exclusion criteria

which may make it more difficult to synthesise or cross compare. Would be helpful to more clearly define your symptoms of interest a priori eg using International Classification of Primary Care (ICPC).

We do not want to be too prescriptive in the definition of symptoms as there may be different classification systems used across different settings and we do not wish to limit the study to a narrow definition of the symptoms. As we have noted on Page 22 Lines 57-58, "All potential coding classification systems will be eligible." The comment about difficulties in synthesis and cross-comparison with such an approach is noted and has been mentioned in the new limitations section (Page 23, Lines 41-44).

3.3

Although bias in the studies may not be an outcome of interest I would like to see how you plan to address the quality of the methods used for developing the codelists, eg by using previously published methods as a benchmark (such as Watson et al 2017 <https://bmjopen.bmj.com/content/7/11/e019637>)

Thankyou for highlighting the issue of quality of the methods used in codelist development and for the citation for the above paper. We had always planned to collect information about the method used to generate the codelist, but have now incorporated an assessment of the quality of that method, using the above reference and others. We have added the line: "The quality of code set development methodology will be scored using methods established in the literature as a benchmark" (Page 23, Lines 24-26).

3.4

Finally I would like to see acknowledgement in the limitations section that using a retrospective review of published codelists inevitably means that codelists produced will be outdated. This is particularly important given migration to SNOMED-CT. As a result researchers should supplement pre-existing codelists with additional searches for any new clinical codes of relevance.

We have acknowledged this limitation in the new limitations section (Page 23, Lines 44-49).

We have also included another data variable to capture (Page 23, Line 22), if reported: what version of the classification system a code set was created for (e.g. Read v2 April 2016).

3.5

Finally please include the PROSPERO registration number.

We have now included the PROSPERO registration number CRD42018100830.

REVIEWER: 4

This protocol reports on the criteria for a systematic review of code lists for respiratory symptoms used in electronic health record research. This is an important first step when designing and developing a study using EHR data and ensures research conducted is reproducible and more comparable between databases.

4.1

The authors have developed an extensive code list search strategy and identified a good range of code list repositories to search for articles not found through their literature search. Another repository

the authors may wish to consider is the LSHTM Data Compass (<http://datacompass.lshtm.ac.uk/>) which provides code lists for research produced by LSHTM and its collaborators

We have amended our list of clinical code repositories to include the LSHTM Data Compass as suggested (Page 22, Line 45) and included a reference to this.

4.2

A minor point, in the EMBASE and Medline search strategy provided in the supplement, the time window for consideration is reported as 1990-2017 whereas in the protocol it is stated as 1990-2018.

To clarify this, we have amended how the dates are reported in the manuscript: using 'December 2017' instead of 'January 2018' (Page 19, Line 23) and (Page 22, Line 60) in order to better match how it is reported in the search strategy.

VERSION 2 – REVIEW

REVIEWER	Mohammad Al Sallakh Swansea University Medical School
REVIEW RETURNED	21-Jan-2019

GENERAL COMMENTS	The authors have satisfactorily addressed my comments.
--

REVIEWER	Susan Fenton The University of Texas School of Biomedical Informatics
REVIEW RETURNED	04-Feb-2019

GENERAL COMMENTS	The title states that this is a systematic review protocol, yet the summary of the article states it is a systematic review. In the methods, the overall objective is a review. Is this a review or a protocol for reviews? The design of the protocol would be strengthened with a draft PRISMA diagram. Without completing the review (as is alluded to in the document) it is hard to judge the quality of the protocol. Other than the search strategy, there are no results or outcomes.
---